# Secure Out-of-Distribution Task Generalization with Energy-Based Models

**Shengzhuang Chen**[1]    **Long-Kai Huang**[2]    **Jonathan Richard Schwarz**[3]
**Yilun Du**[4]    **Ying Wei**[1,5*]
[1]City University of Hong Kong    [2]Tencent AI Lab    [3]University College London
[4]Massachusetts Institute of Technology    [5]Nanyang Technological University
`szchen9-c@my.cityu.edu.hk`    `{hlongkai, schwarzjn}@gmail.com`
`yilundu@mit.edu`    `ying.wei@ntu.edu.sg`

## Abstract

The success of meta-learning on out-of-distribution (OOD) tasks in the wild has proved to be hit-and-miss. To safeguard the generalization capability of the meta-learned prior knowledge to OOD tasks, in particularly safety-critical applications, necessitates detection of an OOD task followed by adaptation of the task towards the prior. Nonetheless, the reliability of estimated uncertainty on OOD tasks by existing Bayesian meta-learning methods is restricted by incomplete coverage of the feature distribution shift and insufficient expressiveness of the meta-learned prior. Besides, they struggle to adapt an OOD task, running parallel to the line of cross-domain task adaptation solutions which are vulnerable to overfitting. To this end, we build a single coherent framework that supports both detection and adaptation of OOD tasks, while remaining compatible with off-the-shelf meta-learning backbones. The proposed Energy-Based Meta-Learning (EBML) framework learns to characterize any arbitrary meta-training task distribution with the composition of two expressive neural-network-based energy functions. We deploy the sum of the two energy functions, being proportional to the joint distribution of a task, as a reliable score for detecting OOD tasks; during meta-testing, we adapt the OOD task to in-distribution tasks by energy minimization. Experiments on four regression and classification datasets demonstrate the effectiveness of our proposal.

## 1 Introduction

*Meta-learning* [48, 6] that builds general-purpose learners with limited data has been under constant investigation, recently demonstrating its potential to even advance few-shot learning of large language models [36, 44]. Analogous to the notorious domain shift [23] that degrades the performance of deep learning, meta-testing tasks that are out of the distribution of meta-training tasks (*a.k.a. out-of-distribution (OOD) tasks*) put the meta-learned prior knowledge at high risk of losing effectiveness [46]. In real-world applications, though, out-of-distribution tasks are highly prevalent, e.g., bin picking for a robot that has never been meta-trained on environments involving bins [55], MRI-based pancreas segmentation given a host of meta-training tasks with pathology images [35], and etc. Thus, it is imperative to secure the generalization ability of the meta-learned prior (i.e., meta-generalization) to OOD tasks, especially in safety-critical applications such as medical image analysis.

The *first* step to securing meta-generalization to a task is to develop awareness of whether the task is OOD or not, i.e., **OOD task detection**. Existing solutions in literature have pursued a variety of Bayesian meta-learning methods [7, 54, 41, 10, 43] that balance between flexibility and tractability of

---

*Correspondence to Ying Wei

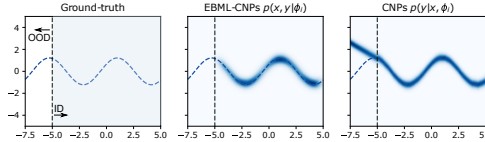

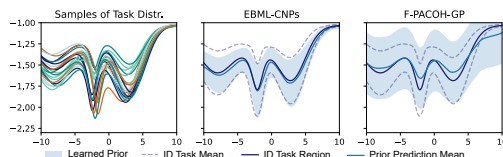

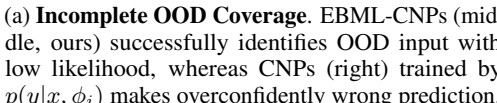

(a) **Incomplete OOD Coverage**. EBML-CNPs (middle, ours) successfully identifies OOD input with low likelihood, whereas CNPs (right) trained by $p(y|x, \phi_i)$ makes overconfidently wrong prediction.

(b) **Limited Expressiveness**. The meta-training task distribution learned by EBML-CNPs (middle, ours) outperforms F-PACOH-GP [43] whose prior distribution is built on GP.

Figure 1: Comparison of EBML and Bayesian meta-learning baselines for OOD detection.

solving the hierarchical probabilistic model $p(\mathbf{Y}_i|\mathbf{X}_i) = \iint p(\mathbf{Y}_i|\mathbf{X}_i, \phi_i)p(\phi_i|\theta)p(\theta)\, d\phi_i d\theta$, where $\mathcal{T}_i = \{\mathbf{X}_i, \mathbf{Y}_i\}$ represents the $i$-th task. $\theta$ and $\phi_i$ denote parameters of the meta-model and task-specific model, respectively. Unfortunately, these methods present some limitations in their practical usage. (1) *Incomplete OOD coverage*: given that the Bayesian uncertainty is trained via maximizing the posterior $p(\mathbf{Y}_i|\mathbf{X}_i)$ above, it is not necessarily high when encountering an OOD task that shares the predictive function $p(\mathbf{Y}_i|\mathbf{X}_i)$ with some meta-training tasks but differs substantially in feature distributions $p(\mathbf{X}_i)$. We verify this in Figure 1a and Appendix D. (2) *Limited expressiveness*: for tractability purpose, the meta-learned prior $p(\phi_i|\theta)$ predicates on simple known distributions, e.g., Maximum A Posterior (MAP) estimation [6, 53] and Gaussian [41, 7, 43], which may struggle to align with the complex probabilistic structure of the meta-training task distribution (see Figure 1b). This misalignment inevitably leads to unreliable estimation of OOD tasks.

Upon detection of an OOD task, *secondly*, adaptation of the meta-learned prior promotes its generalization to this OOD task. We dub this strategy during meta-testing as **OOD task adaptation**, which is closely related to cross-domain meta-learning [4, 25, 34, 49]. The core philosophy behind cross-domain meta-learning is the introduction of task-specific parameters which are inferred via either gradient descent [28, 29] or feed-forward amortized encoder [42, 8] on the support set of each OOD task. Learning task-specific parameters, however, is *prone to overfitting* given the usually very limited size of a support set (e.g., 5 examples only in 5-way 1-shot classification).

The limitations are further complicated by the detachment of the existing solution to OOD task detection from that to OOD task adaptation. An explicit prior model is absent in existing Bayesian meta-learning methods for OOD task detection, so that adapting the prior during meta-testing to accommodate an OOD task is ambitious to achieve. On the other hand, cross-domain meta-learning approaches by design do not offer uncertainty estimation, thereby being a risky OOD task detector. Pursuing a coherent framework that supports both detection and adaptation of OOD tasks remains an open question, which motivates our proposal of a novel probabilistic meta-learning framework.

By virtue of the flexibility and expressiveness of energy-based models [24] in modelling complex data distributions, we propose the Energy-Based Meta-Learning (EBML) framework that overcomes the above-mentioned limitations. Specifically, we derive an energy-based model to explicitly model any meta-training task distribution, resulting in the composition of an explicit prior energy function and a complexity energy function. The sum of the two energy functions, trained directly to meet the joint distribution $p(\mathbf{X}_i, \mathbf{Y}_i)$ and parameterized with neural networks, has *completeness and expressiveness* advantages that give it an edge in detection of OOD tasks. During meta-testing, we iteratively update the parameter for a task that has been identified as OOD by gradient descent of energy minimization, which eventually adapts the prior towards in-distribution tasks and maximally leverages the meta-learned prior for *alleviating overfitting*.

The key contributions of this research are outlined below. (1) *Coherence and generality*: we provide a coherent probabilistic model that allows both detection and adaptation of OOD tasks. Also, EBML is agnostic to meta-learning backbones, being general to secure meta-generalization for arbitrary off-the-shelf meta-learning approaches against OOD tasks. (2) *Practical efficacy*: we conduct our experiments on three regression and one classification datasets, on which EBML outperforms SOTA Bayesian meta-learning methods for OOD task detection with an improvement of up to 7% on AUROC and cross-domain meta-learning approaches for OOD task adaption with up to 1.5% improvement.

## 2 Related Work

**Bayesian Meta-learning** There has been a line of literature on Bayesian meta-learning algorithms with predictive uncertainty estimation for safeguarding safety-critical and few-shot applications. Grant

et al. [11] first recast gradient-based meta-learning as a tractable hierarchical Bayesian inference problem. Much of the subsequent research attempts to solve the problem with various approximations. Assuming a sufficient number of meta-training tasks, almost all works use a point estimate for the initialization [41, 19, 10]. However, estimates of exceptions including [54] rely on SVGD [32] for inference and require significant computation for an ensemble of task-specific weights. Several studies that estimate the uncertainty in task-specific parameters after inner-loop adaptation have explored MAP estimates [e.g. 47], sampling from a neural network [10, 53, 42], and variational inference [41, 7, 43]. The uncertainties considered in these methods are often modelled using isotropic Gaussians which suffer from *limited expressiveness*.

**Meta-learning towards OOD Generalization** Recent cross-domain meta-learning methods [e.g. 25, 4, 49, 34] deal with a distribution shift between meta-training and meta-testing tasks, by typically parameterizing deep networks with a large set of task-agnostic and a small set of task-specific weights that encode shared representations and task-specific representations for the training domains, respectively. The works of [42, 1, 34] augment a shared pre-trained backbone with task-specific FiLM [40] layers whose parameters are estimated through an encoder network conditioned on the task's support set. TSA [28] and URL [29] propose to attach task-specific adaptors in matrix form to the pre-trained backbone at test time, inferring their parameters by gradient descent on the support set for each task *from scratch*. On the other hand, SUR [4] and URT [31] pre-train multiple backbones each for an ID training domain, and meta-learn an attention mechanism to selectively combine the pre-trained representations into task-specific ones for ID and OOD classification. While these methods generally have improved performance in the OOD domains of tasks, they nevertheless are not designed with any explicit mechanism for detecting OOD tasks, i.e., lacking OOD awareness.

**EBMs for OOD Detection** Recently, there has been increasing interest in leveraging EBMs for detecting testing samples that are OOD w.r.t. the training data distribution. Liu et al. [33] directly use the energy score for OOD input detection, while Grathwohl et al. in JEM [12] use gradient norm of the energy function as an alternative OOD score; both yield more superior OOD detection performance than traditional density-based detection methods. There are also a number of works that investigate the OOD detection capability of hybird and latent variable EBMs [38, 14, 13], and more advanced training techniques for improving the density modelling hence OOD detection performance of EBMs [5, 2, 57, 3]. While all aforementioned works focus on the standard supervised and unsupervised learning scenarios, Willette et al. in [52] study OOD detection in meta-learning. However, their work differs from EBML in that (a) EBML aims to detect a meta-testing task that is OOD of the meta-training tasks whereas [52] focuses on detecting a query sample that is OOD of the support samples in a meta-testing task, and (b) EBML explicitly meta-learns the distribution of meta-training tasks via the two proposed EBMs and develops the Energy Sum to flag those high-energy tasks as OOD tasks; while [52] resorts to post-hoc OOD detection via energy scaling (akin to temperature scaling in softmax output) without learning any EBM. Moreover, we offer EBML as a generic and flexible probabilistic meta-learning framework that supports both *detection* and *adaptation* of OOD tasks.

## 3 Preliminaries: Energy-based Models

An energy-based model (EBM) [24] expresses a probability density $p(\mathbf{x})$ for $\mathbf{x} \in \mathbb{R}^D$ as

$$p_\theta(\mathbf{x}) = \frac{\exp(-E_\theta(\mathbf{x}))}{Z(\theta)}, \tag{1}$$

where $E_\theta(\mathbf{x})$ is the energy function parametrized by $\theta$ that maps each point $\mathbf{x}$ in the input space to a scalar value known as the *energy*. $Z(\theta) = \int_{\mathbf{x}} \exp(-E_\theta(\mathbf{x}))d\mathbf{x}$ is the partition function that is a constant w.r.t. the variable $\mathbf{x}$. Training $p_\theta(\mathbf{x})$ to fit some data distribution $p_D(\mathbf{x})$ requires maximizing the log-likelihood $\mathcal{L}(\theta) = \mathbb{E}_{\mathbf{x} \sim p_D(\mathbf{x})}[\log p_\theta(\mathbf{x})]$ w.r.t. $\theta$. Though an intractable integral in $Z_\theta$ is involved in this objective, it is not a concern when computing the gradient [3, 12]

$$\nabla_\theta \mathcal{L} = \mathbb{E}_{\mathbf{x}' \sim p_\theta}[\nabla_\theta E_\theta(\mathbf{x}')] - \mathbb{E}_{\mathbf{x} \sim p_D}[\nabla_\theta E_\theta(\mathbf{x})]. \tag{2}$$

Intuitively, Eqn. (2) encourages $E_\theta$ to assign low energy to the samples from the real data distribution $p_D$ while assigning high energy to those from the model distribution $p_\theta$. Computing Eqn. (2), thus, requires drawing samples from $p_\theta$, which is challenging. Recent approaches [12, 3] on training EBMs resort to stochastic gradient Langevin dynamics (SGLD) [51] which generates samples following

$$\mathbf{x}^0 \sim p_0(\mathbf{x}), \qquad \mathbf{x}^{k+1} = \mathbf{x}^k - \frac{\eta^2}{2} \frac{\partial E_\theta(\mathbf{x}^k)}{\partial \mathbf{x}^k} + \eta \mathbf{z}^k. \tag{3}$$

The $K$-step sampling starts from an (typically uniform) initial distribution $p_0(\mathbf{x})$. $\mathbf{z}^k \sim \mathcal{N}(\mathbf{0}, \mathbf{I}) \in \mathbb{R}^D$ is a perturbation, and $\eta \in \mathbb{R}^+$ controls the step size and noise magnitude. Denote the distribution $q_\theta$ by Eqn. (3), which signifies $\mathbf{x}' = \mathbf{x}^K \sim q_\theta$. When $\eta \to 0$ and $K \to \infty$, then $q_\theta \to p_\theta$ under some regularity conditions [51]. Consequently, the gradient of Eqn. (2) is approximated in practice [3, 12] by

$$\nabla_\theta \mathcal{L} = \mathbb{E}_{\mathbf{x}' \sim \text{stop\_grad}(q_\theta)}[\nabla_\theta E_\theta(\mathbf{x}')] - \mathbb{E}_{\mathbf{x} \sim p_D}[\nabla_\theta E_\theta(\mathbf{x})], \tag{4}$$

where the gradient does not back-propagate into SGLD sampling.

## 4  Energy-Based Meta-learning

For clarity, we use the notation $\mathcal{P}_{ID}$ to denote the unknown meta-training ID task distribution where the $i$-th training task is $\mathcal{T}_i$. We let $\mathbf{X}_i$, $\mathbf{Y}_i$ to denote sets of samples $\{\mathbf{x}_{ij}, y_{ij}\}$ in $\mathcal{T}_i$, and $\mathcal{T}_i^s$, $\mathcal{T}_i^q$ to denote support and query sets, respectively. The size of $\mathcal{T}_i$, $\mathcal{T}_i^s$, $\mathcal{T}_i^q$ is denoted by $N_i$, $N_i^s$, $N_i^q$, respectively. The subscript $i$ denotes the task index, and $j$ denotes the sample index.

### 4.1  Energy-based Modelling of Task Distribution

As illustrated in Introduction, existing probabilistic meta-learning methods maximizing the predictive likelihood $p(\mathbf{Y}|\mathbf{X})$ suffer from incomplete OOD coverage. To this end, we model the meta-training task distribution by (1) formulating the **joint distribution** $p(\mathbf{X}_i, \mathbf{Y}_i)$ of each task $\mathcal{T}_i$ and (2) maximizing the log-likelihood of all meta-training tasks. Concretely, by Kolmogorov's extension and de Finneti's theorems [22], we have the expected log-likelihood of the meta-training tasks as $\mathbb{E}_{\mathcal{P}_{ID}}[\log p(\mathcal{T}_i)] = \mathbb{E}_{\mathcal{P}_{ID}}[\log p(\mathbf{X}_i, \mathbf{Y}_i)] = \mathbb{E}_{\mathcal{P}_{ID}}[\log \int_{\phi_i} \prod_{j=1}^{N_i} p(\mathbf{x}_{ij}, y_{ij}|\phi_i) p(\phi_i) d\phi_i]$. Each $p(\mathcal{T}_i)$ is written in a factorized form over $N_i$ conditional independent distributions with $\phi_i$ being the task-specific latent variable. Due to the intractable integral over $\phi_i$ in high dimension, we resort to amortized inference [8, 41] and learn with a lower-bound instead. This gives the ELBO

$$\mathbb{E}_{\mathcal{P}_{ID}}[\log p(\mathcal{T}_i)] \geq \mathbb{E}\left[\mathbb{E}_{\phi_i \sim q_\psi(\phi_i|\mathcal{T}_i^s)}\left[\log \prod_{j=1}^{N_i} p(\mathbf{x}_{ij}, y_{ij}|\phi_i)\right] - \text{KL}\big(q_\psi(\phi_i|\mathcal{T}_i^s)||p(\phi_i)\big)\right]. \tag{5}$$

Following the conventional wisdom [41, 28, 6], $q_\psi$ is conditioned on the support set only during meta-training to align the inference procedure, i.e., $\phi_i \sim q_\psi(\phi_i|\mathcal{T}_i^s)$, for meta-training and meta-testing. It remains now to parameterize the three distributions in Eqn. (5) including **(a)** the task-specific data distribution $p(\mathbf{x}_{ij}, y_{ij}|\phi_i)$, **(b)** the prior latent distribution $p(\phi_i)$, and **(c)** the posterior latent distribution $q_\psi(\phi_i|\mathcal{T}_i^s)$. Prior works parameterize these distributions in simple known forms, e.g., Gaussians [41, 7, 43] or MAP estimation [6, 53], which may be insufficient to match the complex probabilistic structure of the meta-training task distribution. To increase the expressiveness, we turn to EBMs for parameterizing the two distributions of $p(\mathbf{x}_{ij}, y_{ij}|\phi_i)$ and $p(\phi_i)$. For one reason, EBMs are known to be sufficiently flexible and expressive for characterizing complex arbitrary density functions [3] not limiting to only uni-modal distributions like isotropic Gaussians and MAP estimation; for another, the energy function of an EBM is directly proportional to the negative log-likelihood, paving the way for OOD detection in Section 4.2.

**(a) Task-specific data EBM** We model $p(\mathbf{x}_{ij}, y_{ij}|\phi_i)$ by an energy function parameterized with $\omega$,

$$p(\mathbf{x}_{ij}, y_{ij}|\phi_i) = p_\omega(\mathbf{x}_{ij}, y_{ij}|\phi_i) = \frac{\exp(-E_\omega(\mathbf{x}_{ij}, y_{ij}, \phi_i))}{Z(\omega, \phi_i)}, \tag{6}$$

where $E_\omega$ denotes the task-specific data energy function conditioned on the latent $\phi_i$, and $Z(\omega, \phi_i)$ is the corresponding partition function. Note that the parameter $\omega$ of this EBM is shared by all tasks.

**(b) Latent prior EBM** Inspired by [39], we model the prior latent distribution $p(\phi_i)$ as an unconditional EBM parameterized by $\lambda$; training such a EBM offers expressiveness benefits over a fixed non-informative prior distribution, e.g., isotropic Gaussian distribution. Specifically,

$$p(\phi_i) = p_\lambda(\phi_i) = \frac{\exp(-E_\lambda(\phi_i))}{Z(\lambda)}, \forall i. \tag{7}$$

**(c) Latent posterior** As many meta-learning algorithms have already carefully designated the posterior latent distribution $q_\psi(\phi_i|\mathcal{T}_i^s)$, we simply follow the same implementation of $q_\psi$ in the chosen base meta-learning algorithm, e.g., MAP estimation in [8, 42, 1, 53]. This design favorably empowers EBML to be a generic and flexible framework compatible with off-the-shelf meta-learning algorithms.

Grounded on the above parameterization, we are now ready to derive our EBML **meta-training objective** as below by plugging the two EBMs defined in Eqn. (6) and Eqn. (7) into Eqn. (5). The derivation shares the spirit with Eqn. (4), and more details can be found in Appendix A.1.

$$\underset{\omega,\psi,\lambda}{\arg\max}\, \mathbb{E}_{\mathcal{T}_i\sim\mathcal{P}_{ID}}\bigg[\mathbb{E}_{\boldsymbol{\phi}_i\sim q_\psi(\boldsymbol{\phi}_i|\mathcal{T}_i^s)}\big[\sum_{j=1}^{N_i}-E_\omega(\mathbf{x}_{ij},y_{ij},\boldsymbol{\phi}_i)+\mathbb{E}_{p_\omega(\mathbf{x}',y'|\phi_i)}[E_\omega(\mathbf{x}'_{ij},y'_{ij},\boldsymbol{\phi}_i)]]$$

$$-\mathbb{E}_{q_\psi(\boldsymbol{\phi}_i|\mathcal{T}_i^s)}[E_\lambda(\boldsymbol{\phi}_i)]+\mathbb{E}_{p_\lambda(\boldsymbol{\phi}_i')}[E_\lambda(\boldsymbol{\phi}_i')]+\mathcal{H}(q_\psi(\boldsymbol{\phi}_i|\mathcal{T}_i^s))\bigg]. \qquad (8)$$

Solving the above meta-training objective involves sampling of $\mathbf{x}', y'$ from $p_\omega$ and $\boldsymbol{\phi}_i'$ from $p_\lambda$, in order to compute the expectations $\mathbb{E}_{p_\omega(\mathbf{x}',y'|\phi_i)}$ and $\mathbb{E}_{p_\lambda(\phi_i')}$ as Monte-Carlo averages. We follow the similar SGLD sampling procedure in Eqn. (3). Besides, since the majority of state-of-the-art meta-learning algorithms [8, 42, 1, 53] adopt the MAP estimation of the latent posterior $q_\psi$ which is deterministic, the last entropy term of $\mathcal{H}$ essentially becomes zero and the expectations in the first and second terms are trivial to solve. For this reason, we focus on base meta-learning algorithms with MAP approximation in the following sections, which not only simplifies computation but also maintains the state-of-the-art performance. We left a discussion on EBML with distributional $q_\psi$ in Appendix C.3. The complete pseudo codes for meta-training of EBML are available in Appendix E.

## 4.2   EBML for OOD Detection

Detecting an OOD task w.r.t. the meta-training distribution constitutes an essential first step to guard successful meta-generalization. A straightforward solution is density-based OOD detection, for which the OOD score of a task following the Bayesian principle boils down to its log-likelihood $\log p(\mathbf{X}_i^s,\mathbf{Y}_i^s)=\log\mathbb{E}_{\boldsymbol{\phi}_i\sim p_\lambda(\phi_i)}[p_\omega(\mathbf{X}_i^s,\mathbf{Y}_i^s|\boldsymbol{\phi}_i)]$. Despite the meta-learned latent prior EBM $p_\lambda(\boldsymbol{\phi}_i)$ that is readily available, estimating this log-likelihood still presents daunting challenges. First, when the latent prior is expressed in the form of a distribution over model parameters in very high dimension, MCMC sampling from $p_\lambda(\boldsymbol{\phi}_i)$ is almost computationally infeasible. Second, especially when the latent prior exhibits multi-modality, drawing a considerable number of samples to achieve a low-variance MC estimation of the integral is prohibitively costly.

On this account, we define the OOD score of a task to be faithful to our proposed ELBO approximation of its log-likelihood in Eqn. (5), which gives

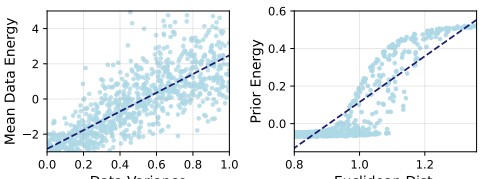

Figure 2: The roles of $E_\omega$ and $E_\lambda$ in Energy Sum in detecting OOD tasks. Each dot denotes a task. **Left**: We perturb each support sample of a task $\mathcal{T}_i$ by $\boldsymbol{\eta}_{ij}\sim\mathcal{N}_i(0,\sigma_i)$ where we sample $\sigma_i$ from $[0,1]$ uniformly. The $y$-axis shows the average energy $\mathbb{E}_{\mathbf{x}_{ij}^s,\mathbf{y}_{ij}^s\sim\mathcal{T}_i^s,\boldsymbol{\eta}_{ij}\sim\mathcal{N}_i}[E_\omega(\mathbf{x}_i^s,\mathbf{y}_i^s,\boldsymbol{\phi}_i)]$ and the $x$-axis plots the variance $\sigma_i^2$. **Right**: We first compute the mean of the overall ID task latent prior as $\boldsymbol{\phi}_{ID}=\mathbb{E}_{\boldsymbol{\phi}_i\sim p_{ID}}[\boldsymbol{\phi}_i]$. The $y$-axis shows the energy $E_\lambda(\boldsymbol{\phi}_{ID}+\boldsymbol{\eta}_i)$ where $\boldsymbol{\eta}_i\sim\mathcal{N}(0,1)$ for the $i$-th task and the $x$-axis plots the Euclidean distance of the perturbed latent from $\boldsymbol{\phi}_{ID}$.

$$\mathbb{E}_{q_\psi(\boldsymbol{\phi}_i|\mathcal{T}_i^s)}\big[\sum_j^{N_i^s}E_\omega(\mathbf{x}_{ij}^s,y_{ij}^s,\boldsymbol{\phi}_i)+E_\lambda(\boldsymbol{\phi}_i)\big]. \qquad (9)$$

We dub this OOD score tailored to EBML **Energy Sum**, whose full derivation is deferred to Appendix A.2. This energy sum enjoys not only the theoretical advantage, i.e., being provably proportional to the negative log-likelihood of a task, but also simple computation benefits. During meta-testing, evaluating the score of Eqn. (9) for each task requires only a single forward pass of the support set samples through the two energy functions.

More remarkably, the energy sum is intuitively appealing in the sense that it characterizes (1) *how far a task is from the overall ID meta-training task distribution* via the latent prior energy score $E_\lambda$ and (2) *how difficult it is to predict the observed support set conditioned on $\boldsymbol{\phi}_i$* via the task-specific data energy score $E_\omega$. First, the terms in the last line of Eqn. (8) for learning the latent prior EBM altogether correspond to maximizing the likelihood $\mathbb{E}_{\mathcal{T}_i\sim p(\mathcal{T})}\mathbb{E}_{q_\psi(\boldsymbol{\phi}_i|\mathcal{T}_i^s)}[\log p_\lambda(\boldsymbol{\phi}_i)]$, which enforces the latent prior energy score $E_\lambda$ to capture the overall ID meta-training distribution. As illustrated in

Figure 2 (right), the further away a task is from the overall ID meta-training distribution measured in Euclidean distance, the larger the energy score $E_\lambda$ is as expected. Second, conditioned on even the ID latent prior $\phi_i$, those tasks with support samples as scattered as possible are especially difficult to predict. These tasks are considered to be OOD, as evidenced in higher values of $E_\omega$ in Figure 2 (left).

## 4.3 EBML for OOD Generalization

The Energy Sum proposed in Section 4.2 develops OOD awareness of a meta-testing task, based on which we differentiate our meta-testing procedures for effective meta-generalization.

**Meta-testing for ID tasks** Given the support set $\mathcal{T}^s$ of a meta-testing task, prediction of the label for its query $\mathbf{x}_j^q$ amounts to maximizing our approximated log-likelihood (see Eqn. (5)) of the task, i.e.,

$$y_j^q = \arg\min_y \mathbb{E}_{\phi \sim q_\psi(\phi \mid \mathcal{T}^s)} \left[ E_\omega(\mathbf{x}_j^q, y, \phi) + E_\lambda(\phi) \right]. \tag{10}$$

Provided that the task has already been identified within the ID region, the second energy $E_\lambda(\phi)$ is negligibly small. Consequently, we reduce the above optimization problem to consider only the first term $E_\omega(\mathbf{x}_j^q, y, \phi)$, and solve it via gradient descent. We provide the pseudo codes in Appendix E.

**Meta-testing for OOD tasks** For an OOD task, its meta-learned prior $\phi \sim q_\psi(\phi \mid \mathcal{T}^s)$ is located out of the ID meta-training task distribution and likely loses its effectiveness. We seek a solution that adapts this inadequate meta-learned prior back to the ID region, so as to make the most of the ID latent priors with guaranteed meta-generalization. This shares the idea with classifier editing in [45], where the editing parameters are trained to map an OOD image to an ID one for improving generalization. Therefore, we introduce task-specific parameters $\zeta$ which are optimized via the following,

$$\arg\min_\zeta \mathbb{E}_{\phi \sim q_{\psi \cup \zeta}(\phi \mid \mathcal{T}^s)} \left[ \sum_{j=1}^{N^s} E_\omega(\mathbf{x}_j^s, y_j^s, \phi) + \max(E_\lambda(\phi) - m, 0) \right], \tag{11}$$

where $m$ is a hyper-parameter. We find that setting $m$ as the empirical average of the latent prior energy over all ID training tasks works well in practice, i.e., $m = \mathbb{E}_{p_{ID}}[\mathbb{E}_{\phi_i \sim q_\psi(\phi_i \mid \mathcal{T}_i^s)}[E_\lambda(\phi_i)]]$.

As a result of optimizing the second term in Eqn. (11), the task-specific parameters $\zeta$ enable $q_{\psi \cup \zeta}(\phi \mid \mathcal{T}^s)$ to accommodate for OOD tasks by mapping the meta-learned prior back to ID meta-training tasks; while optimizing the first term preserves the data-level predictive ability of the model. We highlight that the task energy minimization approximates the minimization of a KL divergence between the task-specific posterior and the meta-learned prior, thereby inducing a meta-regularization effect during adaptation. See Appendix A.3 for details. Eventually, we use the adapted task-specific parameters for final prediction on query samples as in Eqn. (10). Pseudo code for the EBML adaptation and inference algorithms described above can be found in Appendix E.

In Figure 3, we visualize the adaptation process when optimizing Eqn. (11) for OOD few-shot classification tasks in Meta-dataset [49]. As the prior

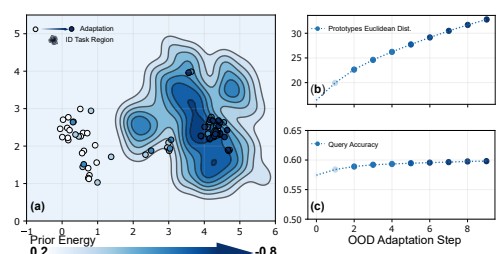

Figure 3: Illustration of the OOD task adaptation process on OOD domains of the meta-dataset [49] where each dot in (a) represents an OOD task in latent space $\phi$. Minimizing Eqn. (11) leads to **(a)** the latent $\phi$ of the OOD task moving to the ID region (contour plot), **(b)** the Euclidean distance between class prototypes enlarging, and consequently **(c)**, the classification accuracy on query samples increasing.

energy of these OOD tasks decreases, their $\phi_i$ gradually shift towards to the ID region as desired. Within this region, minimizing the first term in Eqn. (11) continuously improves generalization. In contrast, given only a few support samples, existing SOTA methods that solely rely on feed-forward inference [1] and gradient-based optimization [28] for OOD task adaptation without a prior are both prone to overfitting. We provide more empirical evidence on this in Appendix C. On the other hand, meta-learning a BNN, which imposes a prior distribution on the parameter space during adaptation may be computationally cumbersome and often lead to sub-optimal performance in comparison to their non-Bayesian counterparts.

# 5 Experiments

In the experiments, we test EBML on both few-shot regression and image classification tasks in search for answers to the following key questions: **RQ1:** Whether the improved expressiveness of EBML over traditional Bayesian meta-learning methods can lead to a more accurate model of the meta-training ID task distribution, hence a more reliable OOD task detector. **RQ2:** Whether Energy Sum can be an effective score for detection of OOD meta-testing tasks. **RQ3:** Whether EBML instantiated with SOTA algorithms can exploit the meta-learned EBM prior in OOD task adaptation to achieve better prediction performance on OOD tasks.

## 5.1 Implementation Details

We now discuss two instantiations of the EBML framework with SOTA meta-learning algorithms for regression and classification. We illustrate our approach in Figure 4 below and defer a more detailed description for our models to Appendix B.

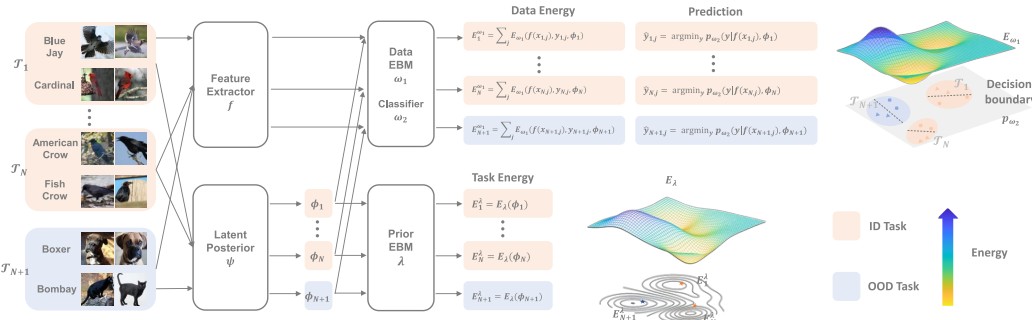

Figure 4: Overview of the EBML framework. The task latent variable $\phi_i$ is inferred from the support set $\mathcal{T}_i^s$ following the implementation of the base algorithm. The data and task energy scores are evaluated by the data and prior EBMs $E_{\omega_1}$ and $E_\lambda$, respectively; while the query labels are predicted by the classifier $p_{\omega_2}$ of the base algorithm.

**Regression.** Take CNPs [8] as an example base model. CNPs implements $q_\psi(\phi_i | \mathcal{T}_i^s)$ as a neural network encoder that outputs a function embedding in finite vector form, i.e., $\phi_i \in \mathbb{R}^D$, from a given support set, $\mathcal{T}_i^s$. That said, we let the prior EBM to model the empirical distribution over such finite-dimension function embedding, i.e., $E_\lambda(\phi_i) : \mathbb{R}^D \to \mathbb{R}$.

**Classification** Many cross-domain few-shot classification algorithms [28, 42, 1] rely on a metric-based classifier for prediction, which assigns query sample to the class with nearest prototype to the query representation based on some distance measure. In these cases, it is natural to specify the task-specific latent $\phi_i$ as the set of class prototypes in each ID training task. Since $\phi_i$ is a set of variables, we build the prior EBM model as a permutation-invariant neural network function. Suitable choices include DeepSets [56] and set transformer [26].

To align with the state-of-the-art prediction performance, we follow the practice in [50, 37] to train another decoder $\omega_2$ with the loss function (e.g., cross entropy) in the base meta-learning model, which serves as a surrogate for $E_\omega(x_j^q, y, \phi)$ in Eqn. (10) and Eqn. (11). We use this decoder for prediction.

**Baseline Models** For regression, we compare against: 1) MAML [6] which is a deterministic meta-learning method, and 2) Bayesian meta-learning methods that use Gaussians for prediction or prior, including ABML [41], MetaFun [53], CNPs [8] and F-PACOH-GP [43]. For classification, we consider Simple-CNAPs [1] and TSA [28], which respectively resort to amortized variational inference and gradient-based optimization for estimating the task-specific parameters from the support set. Both are SOTA cross-domain few-shot classification approach on the Meta-dataset [49] benchmark. For more experimental details, hyper-parameter configurations, and additional experimetal results, please refer to Appendix B and C.

## 5.2 Datasets and Evaluation Metrics

**Sinusoids Few-shot Regression** We consider 1D sinusoids regression tasks in the form $y(x) = A\sin(B(x+C))$. For ID meta-training, we consider frequency $B = 1$, while sample amplitude $A$ and phase $C$ uniformly from a set of equally-spaced points $\{1, 1.1, 1.2, ..., 4\}$ and $\{0, 0.1, 0.2, ..., 0.5\pi\}$, respectively. Each training task consists of 2 to 5 support and 10 query points with $x$ uniformly sampled from $\mathcal{X} \in [-5.0, 5.0]$. During testing, we evaluate the models on 500 ID and OOD tasks each with 512 equal-distant query points in $\mathcal{X}$. For ID testing, we expand the range of the tasks by uniformly sampling $A \in [1, 4]$ and $C \in [0, 0.5\pi]$. For OOD tasks, we randomly change either the phase distribution to $C \in [0.6\pi, 0.75\pi]$, amplitude to $A \in [0.1, 0.8] \cup [4.2, 5.0]$ or frequency to $B \in [1.1, 1.25]$. Details for the multi-sinusoids regression experiment can be found in C.1. We use **MSE** and negative log-likelihood on query samples to evaluate the regression performance.

**Drug Activity Prediction Few-shot Regression** In each task, we aim to predict the drug-target binding affinity of query molecular compounds given 10 to 50 labelled examples from the same domain defined by molecular size. We use the *lbap-general-ic50-size* ID/OOD task split in the DrugOOD [21] benchmark, which divides the molecules into 222/145/23 domains by molecular size for ID Train / ID Test / OOD Test, respectively. The regression performance is evaluated by the square of Pearson coefficient ($\mathbf{R^2}$) between predictions and the ground-truth values. We report the mean and median $R^2$ on 500 tasks sampled from ID and OOD testing domains.

**Meta-dataset [49] 5-way 1-shot Classification** This experiment considers image classification problems on Meta-dataset [49]. Each task contains up to 10 query images per class from the same domain. Following the current state-of-the-art practice [28, 1], we use Aircraft, dtd, cub, vgg-flower, fungi, quickdraw and omniglot as the ID datasets for meta-training and meta-testing, while traffic, mscoco, cifar10, cifar100 and mnist are treated as OOD datasets for meta-testing only.

**OOD Task Detection Evaluation** We compare the OOD task detection performance of Energy Sum against several model-agnostic OOD detection baselines. Concretely, for classification, we compare against max-softmax score [16], ODIN [30], MAH [27], and max-logits score [15]; for regression, we consider Averaged Bayesian prediction uncertainty in standard deviation (**Std**) on support samples, and Averaged **S**upport samples **N**egative **L**og-**L**ikelihood (**SNLL**) under model's task-specific predictive probability, i.e., $-\mathbb{E}_{\phi_i \sim q_\psi(\phi_i | \mathcal{T}_i^s)}[\mathbb{E}_j[\log p_\omega(y_{ij}^s | x_{ij}^s, \phi_i)]]$ for baselines and $\mathbb{E}_{\phi_i \sim q_\psi(\phi_i | \mathcal{T}_i^s)}[\mathbb{E}_j[E_\omega(y_{ij}^s, x_{ij}^s, \phi_i)]]$ for EBML. Following common practice [17, 16], we report **AUROC**, **AUPR** and **FPR95** for OOD detection performance. Details for these metrics can be found in Appendix B.1.

## 5.3 OOD Detection Results

**Energy sum performs best in OOD task detection.** Table 1 and 8. The proposed energy sum further improves our SNLL-only results in all three OOD detection metrics - with 15.2% and 11.8% significant reduction in FPR95, outperforming the best baseline methods by 20.0% and 39.1%, in single and multi-sinusoids situations respectively. In Table 2 for OOD classification task detection, Energy Sum consistently results in superior OOD detection performance, outperforming the best baselines by large margins of 36.84% and 20.19% in FPR95 for Simple-CNAPs and TSA, respectively.

Table 1: OOD task detection performance on single-sine and DrugOOD [21] few-shot regression tasks.

| OOD Scores | Models | Sinusoids | | | DrugOOD | | |
|---|---|---|---|---|---|---|---|
| | | AUROC↑ | AUPR↑ | FPR95↓ | AUROC↑ | AUPR↑ | FPR95↓ |
| Std | ABML [41] | 50.14 | 54.80 | 97.20 | 57.82 | 50.31 | 74.80 |
| | F-PACOH-GP [43] | 49.52 | 51.30 | 94.20 | 81.74 | 71.99 | 32.00 |
| | CNPs [8] | 22.72 | 35.34 | 99.60 | 93.56 | 89.58 | 13.00 |
| | Metafun [53] | 76.57 | 80.33 | 82.40 | 85.68 | 80.55 | 58.18 |
| SNLL | ABML [41] | 82.48 | 81.31 | 61.00 | 80.99 | 79.12 | 47.60 |
| | F-PACOH-GP [43] | 91.78 | 93.23 | 52.40 | 37.73 | 45.01 | 85.21 |
| | CNPs [8] | 95.63 | 96.46 | 34.22 | 17.25 | 34.07 | 91.40 |
| | Metafun [53] | 96.25 | 97.11 | 32.00 | 83.54 | 85.54 | 65.17 |
| | EBML-CNPs (Ours) | 96.46 | 97.41 | 29.40 | 99.71 | 99.71 | 2.20 |
| Energy Sum | EBML-CNPs (Ours) | **97.74** | **98.31** | **14.20** | **99.79** | **99.78** | **1.40** |

**Modelling the joint distribution improves OOD detection under Domain-shift.** In Table 1 DrugOOD regression tasks, using either our SNLL or Energy Sum as OOD scores can achieve better detection performance than baselines. In particular, our method outperforms the best OOD detection results obtained using Gaussian SNLL and Std by 43.84% and 11.6% in FPR95, respectively.

**Qualitative Illustration.** In Figure 5, we visualize the predictive distribution $p(y_{ij}|\mathbf{x}_{ij}, \phi_i)$ learned using an EBM decoder and a Gaussian decoder on a sampled ID multi-sinusoids task. The EBM clearly shows two prediction modes at all non-overlapping positions, whereas the Gaussian decoder is unable to model the multi-modality, resulting in a blurry prediction.

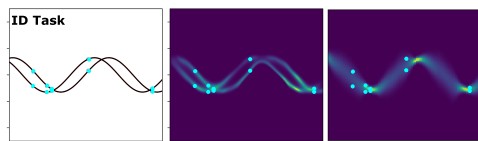

Figure 5: Predictive distribution of **Middle** an data EBM vs **Right** a Gaussian for an ID task.

**Computational Complexity Analysis.** We conduct a computational complexity analysis for EBML by comparing its wall-clock training time and convergence to baselines in Figure 6 below. EBML-CNPs eventually achieves better OOD detection performance than baseline CNPs meanwhile matching its regression performance at all training epochs. In Table 15 Appendix C.4, we show EBML-CNPs is computationally cheaper and faster than traditional Bayesian methods, namely, F-PACOH-GP [43] which requires matrix inversion for inference with Gaussian processes prior, and ABML [41] which imposes a Gaussian prior over the entire parameter space of the model.

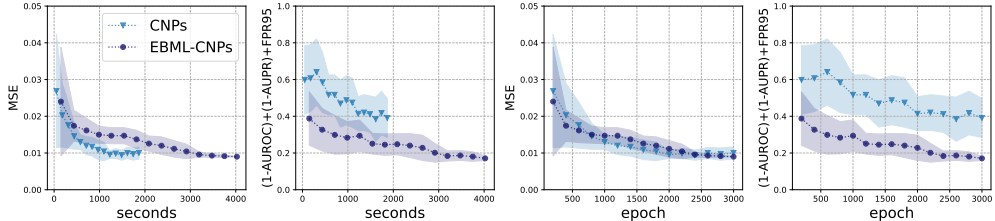

Figure 6: **Left** : Wall-clock convergence in seconds, and **Right**: performance vs number of training epochs, for EBML-CNPs vs CNPs in single-sinosoid few-shot regression tasks. The plots show the regression (MSE ↓) and combined OOD tasks detection (1-AUROC)+(1-AUPR)+FPR95 ↓ performance on single sine few-shot regression tasks during training. Curves are moving averages with window size 3. EBML-CNPs achieves better final performance than CNPs.

**Energy sum achieves better OOD detection results with EMB prior than Gaussian.** In Table 3 and 9, we investigate the contribution of the prior EBM in improving the modelling of meta-training task distribution. We train CNPs and ABML using diagonal Gaussian distribution as the prior in ELBO, and compute OOD scores as **(a)** SNLL, and **(b)** the sum between SNLL and the NLL of task-specific latent evaluated under the learned Gaussian prior (indicated by *+Gauss Prior*). The results show that energy sum using an EBM prior outperforms all ablated models. The OOD detection performance of our model benefits from adding the prior EBM energy to the data EBM energy (SNLL), resulting in the most reduction in FPR95 on both single and multi-sinusoids tasks (15.2% and 11.8%, respectively). This suggests the improved expressive of EBM over simple distributions can indeed lead to learning a more accurate model of the meta-training ID task distribution.

**Energy sum achieves better OOD detection results when learning the joint distribution** In Table 16, we compare EBML-joint, which is exactly our proposed training procedure in the paper,

Table 2: OOD task detection performance on Meta-dataset 5-way 1-shot classification tasks.

| OOD Scores | Simple-CNAPs [1] | | | TSA [28] | | |
|---|---|---|---|---|---|---|
| | AUROC↑ | AUPR↑ | FPR95↓ | AUROC↑ | AUPR↑ | FPR95↓ |
| max-softmax [16] | 85.50 | 85.54 | 65.43 | 89.25 | 87.14 | 46.02 |
| max-logits [15] | 50.00 | 70.83 | 95.00 | 50.14 | 44.64 | 95.28 |
| ODIN [30] | 90.49 | 89.42 | 43.57 | 92.02 | 90.18 | 37.36 |
| MAH [27] | 71.18 | 69.76 | 90.52 | 94.54 | 93.95 | 23.83 |
| Domain Classifier | 83.10 | 73.18 | 53.17 | n/a | n/a | n/a |
| EBML Energy Sum | **97.01** | **94.92** | **6.74** | **99.10** | **98.48** | **3.64** |

Table 3: Ablation study on Energy Sum for OOD detection on single-sinusoids.

| Models | OOD Scores | Sinusoids | | |
|---|---|---|---|---|
| | | AUROC↑ | AUPR↑ | FPR95↓ |
| ABML [41] | SNLL | 82.48 | 81.31 | 61.00 |
| | +Gauss Prior | 86.95 | 86.64 | 52.20 |
| CNPs [8] | SNLL | 94.81 | 96.34 | 38.40 |
| | +Gauss Prior | 94.61 | 96.10 | 34.40 |
| EBML-CNPs | SNLL | 96.46 | 97.41 | 29.40 |
| | +EBM Prior | **97.74** | **98.31** | **14.20** |

Table 4: Few-shot regression performance on single-sinusoids and DrugOOD [21] tasks.

| Models | Sinusoids | DrugOOD | | | |
|---|---|---|---|---|---|
| | ID MSE ↓ | ID Mean $R^2$ ↑ | ID Median $R^2$ ↑ | OOD Mean $R^2$ ↑ | OOD Median $R^2$ ↑ |
| F-PACOH-GP [43] | $0.068_{\pm 0.016}$ | 0.492 | 0.454 | 0.055 | 0.027 |
| Metafun[53] | $\mathbf{0.009}_{\pm 0.002}$ | 0.537 | 0.541 | 0.054 | 0.027 |
| CNPs [8] | $\mathbf{0.009}_{\pm 0.002}$ | **0.540** | 0.549 | 0.066 | **0.046** |
| ABML [41] | $0.127_{\pm 0.013}$ | 0.452 | 0.443 | 0.051 | 0.029 |
| MAML [6] | $0.119_{\pm 0.013}$ | 0.462 | 0.475 | 0.055 | 0.024 |
| EBML-CNPs | $\mathbf{0.009}_{\pm 0.002}$ | 0.533 | **0.553** | **0.071** | 0.043 |

and EBML-conditional, which follows the same training with EBML-joint but models $p(\mathbf{Y} \mid \mathbf{X})$ instead of $p(\mathbf{X}, \mathbf{Y})$. With all other factors being the same, EBML-joint significantly outperform EBML-conditional in OOD detection on DrugOOD regression tasks with domain shift in $\mathbf{X}$. This supports our motivation for using the joint distribution instead of the conditional distribution for training a potentially better OOD detector. Detail of this ablation study can be found in Appendix D.

## 5.4 OOD Generalization Results

**EBML achieves SOTA regression performance.** In Table 4, for single-sinusoids, EBML is able to match the MSE of the best-performing baseline methods; while on multi-sinusoids in Table 7, EBML obtains the lowest ID NLL, specifically 0.58 lower than the best baseline, thanks to our energy-based decoder which is sufficiently expressive for modelling the multi-modality at each input.

**Task adaption using Eqn. (11) improves few-shot classification performance.** In Table 5, we report the average classification accuracy computed over 600 test tasks per ID and OOD domains. In meta-testing, we obtain classification results for EBML-TSA by running gradient descent on the objective in Eqn. (11) to optimize the task-specific modules in TSA from scratch. With this addition of prior energy in the OOD adaption objective, EBML-TSA further improves TSA results in 5/7 ID domains and all 5 OOD domains. Additional OOD classification results in Table 11 Appendix C further confirm the superiority of our proposed OOD task adaptation strategy in Eqn. (11) over prior baselines.

Table 5: Classification performance on 5-way 1-shot tasks for both ID and OOD domains in Meta-dataset.

| Datasets | TSA [28] | EBML-TSA (Ours) |
|---|---|---|
| Omniglot | $98.63_{\pm 0.26}$ | $\mathbf{98.67}_{\pm 0.26}$ |
| Textures | $51.93_{\pm 0.87}$ | $\mathbf{52.35}_{\pm 0.88}$ |
| Aircraft | $\mathbf{78.91}_{\pm 0.86}$ | $78.47_{\pm 0.86}$ |
| Birds | $75.02_{\pm 0.90}$ | $\mathbf{75.52}_{\pm 0.90}$ |
| VGG Flower | $\mathbf{80.37}_{\pm 0.80}$ | $80.30_{\pm 0.83}$ |
| Fungi | $70.89_{\pm 0.93}$ | $\mathbf{72.29}_{\pm 0.94}$ |
| Quickdraw | $79.02_{\pm 0.84}$ | $\mathbf{80.27}_{\pm 0.85}$ |
| MSCOCO | $52.28_{\pm 0.94}$ | $\mathbf{53.03}_{\pm 0.97}$ |
| Traffic Sign | $57.40_{\pm 0.94}$ | $\mathbf{58.85}_{\pm 1.01}$ |
| CIFAR10 | $49.16_{\pm 0.82}$ | $\mathbf{50.04}_{\pm 0.89}$ |
| CIFAR100 | $62.25_{\pm 1.01}$ | $\mathbf{62.77}_{\pm 1.05}$ |
| MNIST | $74.72_{\pm 0.83}$ | $\mathbf{76.08}_{\pm 0.88}$ |
| Avg ID | 76.40 | **76.84** |
| Avg OOD | 59.16 | **60.15** |
| Avg All | 69.22 | **69.89** |

## 6 Conclusion and Limitation

This paper proposes a new energy-based meta-learning (EBML) framework for the first time, which directly characterizes any arbitrary meta-training task distribution using two data and prior energy functions. EBML is compatible with many existing SOTA meta-learning algorithms and allows both detection and adaption of OOD tasks. The sum of the two learned energy functions gives an unnormalized probability distribution proportional to the underlying task likelihood, deployable as OOD scores. The experiment results show the superiority of Energy Sum over traditional methods in detecting both OOD regression and classification tasks, and the possibility of achieving improved OOD adaptation performance with EBML through minimizing the task energy. One **limitation** of EBML is that our current OOD task adaptation strategy does not consider the effect of negative transfer, as some OOD tasks may benefit from adaptating from scratch without ID energy prior regularization. Thus, in future works, we are interested in designing task-specific adaptation strategies for EBML that can selectively adapt OOD tasks for better performance.

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
