# OpenReview forum: "Secure Out-of-Distribution Task Generalization with Energy-Based Models"
_NeurIPS.cc/2023/Conference — NeurIPS 2023 poster_

### Official Review · Reviewer_aM9D · 2023-07-05

**Soundness:** 3 good
**Presentation:** 3 good
**Contribution:** 3 good
**Rating:** 6
**Confidence:** 4

**Summary:**

This paper studies the intersection of OOD generalization and meta-learning, which is rather new. The main claim is that existing meta-learning algorithms may fail to generalize well in OOD settings since they are not specifically designed to solve OOD tasks. To this end, authors proposed a general framework to incorporate OOD into the Bayesian meta-learning framework and use the derived ELBO to solve the problem. Experiments on Sinusoids, DrugOOD, and meta-dataset show the effectiveness of this approach.

**Strengths:**

1. The problem formulation is new and interesting.
2. The proposed framework is general and can solve both ID and OOD tasks.
3. Experiments are supporting the algorithms.

**Weaknesses:**

1. A lack of theoretical analysis using the derived Beyesian framework. I assume a generalization bound can further help understand the superiority of the approach.
2. There are no experiments related to efficiency, which is demonstrated in abstract section. Additionally, I wonder what is the performance and efficiency of the algorithm in larger datasets.
3. ELBO is difficult to optimize. Are there any experiments on hyperparameter choices?

**Questions:**

See weakness

**Limitations:**

See weakness

---

> ### Author Rebuttal · Authors · 2023-08-10
>
> We sincerely appreciate your constructive comments to improve our paper. We detail our response below point by point. Please kindly let us know if our response addresses the questions you had for this paper.
>
> ##### A lack of theoretical analysis using the derived Bayesian framework. I assume a generalization bound can further help understand the superiority of the approach.
> > - We will definitely include a rigorously derived generalization bound in the final version of our manuscript. For now we borrow the meta-generalization bound from Lampert et.al. [r1] for an intuitive explanation.
> >  - We would first recall Lampert's generalization bound for meta-learning, which consists of three parts: (1) the empirical error $\hat{\text{er}}(\mathcal{Q})$, (2) the KL divergence between the hyper-posterior and the hyper-prior $\text{KL}(\mathcal{Q}||\mathcal{P})$, and (3) the expected KL divergence between each task-specific posterior and priors sampled from the hyper-posterior $\mathbb{E}_{P\sim\mathcal{Q}}[\text{KL}(Q_i(S_i,P)||P)]$.
> >
> > - Our argument is that EBML with more flexibility learns a **better hyper-posterior distribution $\mathcal{Q}$ that approximates the ID task distribution more accurately** than simple distributions like Gaussians. As a result, we expect a **lower KL divergence $\mathbb{E}_{P\sim\mathcal{Q}}[\text{KL}(Q_i(S_i,P)||P)]$** between samples of this hyper-posterior and any task-specific posteriors in the ID task distribution (i.e., the above part (3)).
> >
> > [r1] Pentina, A., & Lampert, C.H . A PAC-Bayesian bound for Lifelong Learning. ICML 2014
>
> ##### There are no experiments related to efficiency, which is demonstrated in abstract section. Additionally, I wonder what is the performance and efficiency of the algorithm in larger datasets.
> > - In Line 18 Abstract, we claim **"effectiveness of our approach"** which we have empirically verified:
> >
> >   - In Table 1,2&3, the purposed energy sum consistently outperforms traditional OOD detection baselines in detecting both regression and classification OOD meta-testing tasks.
> >   - In Table 4 and Table r1 in our global response, our proposed OOD adaptation strategy outperforms the baseline method under few-shot classification settings on 5/5 , 4/5 and 5/5 OOD testing domains in the Meta-dataset benchmark for EBML-TSA,  EBML-URL and EBML-SimpleCNAPs, respectively.
> >
> > - In terms of **efficiency on larger datasets**, we have conducted our experiment on the **Meta-dataset benchmark, which is the largest benchmark currently** and has been extensively studied in many meta-learning algorithms, e.g., TSA, SimpleCNAPs.
> >   - We think that EBML shows a reasonable performance vs computational complexity trade-off given that we have seen a noticeable gain in performance.
> > - Additional computational complexity analysis and wall-clock convergence plots can be found in section C.4 of the supplementary material as well as Fig. r1 in the PDF file attached to our global response to reviewers.
>
> ##### Are there any experiments on hyperparameter choices?
> > - We follow the reviewer's suggestion and conduct a sensitivity analysis of EBML's prediction and OOD detection performance w.r.t. a number of important hyperparameters in Fig. r2 in the PDF file attached to our global response.
> >
> >  - For each plot, we vary one of the hyperparameters from its optimal value while keeping the rest unchanged.
> > -  Based on the results, we observe that EBML's performance is quite stable within the region near the optimal hyperparameter values.

---

### Official Review · Reviewer_qEmy · 2023-07-07

**Soundness:** 3 good
**Presentation:** 2 fair
**Contribution:** 3 good
**Rating:** 5
**Confidence:** 2

**Summary:**

In this paper, the authors address the generalization problem of meta-learning methods on out-of-distribution tasks in the wild. However, existing Bayesian meta-learning methods suffer from incomplete convergence of the feature distribution shifts and insufficient expressiveness of meta-learning priors. The authors propose an energy-based meta-learning framework to represent task distributions.

**Strengths:**

A plug-in and effective module EBML is proposed to improve the performance of existing meta-learning methods and achieve the new state of the art. Extensive experiments on four regression and classification datasets demonstrate the effectiveness of the method.

**Weaknesses:**

Some parts of the paper are not very clear, and it takes a long time for the reviewer to understand. These symbols should be prior definited to help reader to fast find the usefull infromation.

**Questions:**

1. In lines 34-35,  \theta and \phi denote the parameters of the meta model and task-specific model. However, some meta-learning methods, such as MAML, use the same network architecture for both the meta-model and the task-specific model. In this paper, \theta and \phi are different networks?
2. Why do meta-testing for ID and OOD tasks have different optimization functions? Can the authors explain the essential difference between Equations (10) and (11)?
3. In Table 4, I tend to see more experimental results in meta-datasets with different baselines and methods.
4. The author lacks some important related work, using energy methods to solve the problem of meta-learning and few-shot learning [1, 2, 3].

[1] Meta-Learning Deep Energy-Based Memory Models.
[2]  Energy-Efficient and Federated Meta-Learning via Projected Stochastic Gradient Ascent
[3] Unsupervised Meta-Learning via Latent Space Energy-based Model of Symbol Vector Coupling

**Limitations:**

see weaknesses and questions

---

> ### Author Rebuttal · Authors · 2023-08-10
>
> Thank you a lot for the constructive comments. You may find our corresponding explanations below for your concerns. We would really appreciate it if you could let us know if you have any further concerns.
> ##### Q1: On what $\theta$ and $\phi$ represent in EBML
> > Lines 33 in the Introduction is a hierarchical probabilistic model [8, 44] that subsumes almost all meta-learning algorithms, where we use $\theta$ and $\phi$ to denote the parameters of the meta-model and task-specific model. Depending on the meta-learning algorithms, $\theta$ and $\phi$ have different implementations.
> > - For gradient-based meta-learning algorithms including MAML, each task-specific-model $\phi$ is obtained from $\theta$ via gradient descent. Thus, $\theta$ and $\phi$ share the same network architecture but differ in parameters.
> > - For amortized meta-learning algorithms including CNPs that we experimented with EBML, $\theta$ include both the parameters for the amortization network and those for the posterior predictive network (i.e., the decoder); $\phi$ is rather an encoded representation of a task in the form of **a finite-dimensional vector** which is used to condition the posterior prediction.
> > - In EBML, we can collect all the meta-learned parameters that are shared by all tasks into $\theta$, including $\omega$ for the task-specific data EBM, $\psi$ for the amortization network, and the latent prior EBM $\lambda$. $\phi\sim q_\psi(\phi|\mathcal{T}^s)$ following amortized meta-learning algorithms denotes a task-specific vector that conditions the task-specific data EBM $E_\omega(\mathbf{x},y,\phi)$.
> ##### Q2 : Why different optimization functions for ID and OOD tasks, i.e., the difference between Eqn. (10) and Eqn. (11)
> > - We first clarify that Eqn. (10) is used for making **predictions** whereas Eqn. (11) is employed for OOD **task adaptation**, thus the optimization is w.r.t. $y$ (the predicted label) in Eqn. (10) but $\zeta$ (the task-specific parameters for adaptation) in Eqn. (11).
> > - Secondly, the meta-testing procedures are indeed different for ID and OOD tasks:
> >   - For ID tasks, we **solve Eqn. (10) directly** where we use the meta-learned amortization network parameters $\psi$ shared by all tasks;
> >   - For OOD tasks, as we have detailed in Line 235-238, we **first solve Eqn. (11)** for adaptation of the amortization network parameters from $\psi$  to $\psi\cup\zeta$, and **then solve Eqn. (10)** where we use the adapted task-specific parameters $\psi\cup\zeta$.
> > - Third, Line 219-223 illustrates **the reason why we have the above difference during meta-testing, i.e., an extra OOD task adaptation step for OOD tasks**. When given an OOD task, the meta-learned prior $\phi\sim q_\psi(\phi|\mathcal{T}^s)$ is located out of the ID meta-training task distribution which is likely to lose its effectiveness. To alleviate this,  we introduce and optimize the task-specific parameters $\zeta$ via minimizing the latent prior energy $E_{\lambda}(\phi)$, so that the adapted parameters $\psi\cup\zeta$ consequently map this inadequate $\phi$ back to the ID $\phi$ region where the model enjoys guaranteed generalization through meta-training on ID tasks.
> > - Finally, Fig. 3 corroborates that such adaptation for OOD tasks indeed brings their $\phi$ back to ID regions and thereby improves the classification accuracy.
> ##### Q3 : More experiment results for Meta-dataset.
> > We will include URL [31] (which is currently the official second-best method on Meta-dataset; note that in our experiments we have already included TSA [30] which is the best) and EBML-URL as additional baselines and experimental results in Table 4.
> > - For URL we use the official code which is publicly available. To implement EBML-URL for task-specific adaptation, we optimize Eqn. (11) w.r.t. to the task-specific feature projection matrix in URL.
> > - We show the results of URL vs. EBML-URL, together with SimpleCNAPs vs EBML-SimpleCNAPs, **in Table r1 in the PDF file attached to our global response**, from which we observe that EBML-URL outperforms URL on 4/5, and EBML-SimpleCNPAs outperforms SimpleCNAPs 5/5 OOD datasets and conclude with the efficacy of the proposed EBML.
> ##### Q4 : Related works on using energy methods to solve the problem of meta-learning and few-shot learning [1, 2, 3].
> > We very much appreciate the suggestion made by the reviewer to include [1,2,3] as related works. However, after some careful analysis, we think [1,2,3] are less relevant to EBML:
> > - In [1], the authors employed **EBMs as an associative memory model** - a system which is able to retrieve a remembered pattern based on its distorted or incomplete version. Based on this, the author proposed to accelerate the reading/writing to the associative memory-model via meta-learning.
> > - In [2], the authors proposed a federated meta-learning algorithm, whose primary objective is to learn a meta-model that can be fine-tuned to a new task both with a few number of samples in a distributed setting and **at low computation and communication energy consumption**.
> > - In [3], the authors studied **unsupervised meta-learning**, and **formulated a top-down generative model where a latent EBM is used to model the latent clusters within a task**. The EBM in [3], therefore, serves a different purpose to the two novel EBMs in EBML which are used for jointly characterizing the ID meta-training task distribution.
> > - In summary, despite meta-learning or (and) EBMs are studied in [1,2,3], **all of them bear very different goals from EBML**. The goal of EBML is to develop a coherent meta-learning framework that enables both OOD task detection and adaptation in meta-testing.
> >
> > [1] Meta-Learning Deep Energy-Based Memory Models, ICLR 2020
> > [2] Energy-Efficient and Federated Meta-Learning via Projected Stochastic Gradient Ascent, GLOBECOM 2021
> > [3] Unsupervised Meta-Learning via Latent Space Energy-based Model of Symbol Vector Coupling, NeurIPS MetaLearn Workshop 2021

---

> > ### Comment · Reviewer_qEmy · 2023-08-21
> > **Response to authors**
> >
> > Dear authors,
> >
> > I appreciate all the clarifications you made during the rebuttal period. However, I still have a question about why the proposed EBML can achieve good performance on both ID tasks and OOD tasks at the same time?

---

> > > ### Author Response · Authors · 2023-08-21
> > >
> > > We thank the reviewer for your reply. Please kindly find below our response on why EBML can achieve good ID and OOD performance at the same time.
> > >
> > > - Why EBML achieves good OOD performance
> > >
> > >   - We propose **an adaptation procedure** shown in Eqn. (11), where we introduce the additional parameters $\zeta$ specific to an OOD task that adapt the OOD prior $\phi\sim q_{\psi\cup\zeta}(\phi\mid\mathcal{T}^s)$. Such adaptation, as shown in Figure 3, successfully **brings OOD priors into the ID region where meta-generalization is guaranteed due to meta-training on ID tasks**.
> > >
> > >   - We provide an analogy with classifier editing [48] in Line 222, in which the classifier is edited/adapted to accommodate an OOD image (a car with wooden tire) so that **the predicted class of the OOD image aligns with an ID one (a car with rubber tire)**.
> > >
> > > - Why EBML maintains good ID performance
> > >
> > >   - The OOD adaptation in Eqn. (11) is **w.r.t. only the additional parameters that are specific to each OOD task**, thereby **having no effect on any ID task that still uses meta-trained parameters**.
> > >
> > >   - Still, use [48] as an analogy. As shown in Figure 5, the performance of the proposed editing method on ID classes (other classes instead of target class in Figure 5) is **still high, as the parameters for ID class prediction do not change at all**.

---

### Official Review · Reviewer_92KR · 2023-07-09

**Soundness:** 3 good
**Presentation:** 3 good
**Contribution:** 3 good
**Rating:** 7
**Confidence:** 5

**Summary:**

This paper proposes EBML, an energy-based meta-learning which models the joint P(X,Y) with two energy functions - one for E(X,Y,phi) that models task-specific joint P(X_i,Y_i|phi) and E(phi) that models task-specific latents P(phi). The motivation is twofold: 1. completeness: energy-based models can naturally distinguish between in and out-of-distributions because it can easily model the joint P(X,Y), 2. expressiveness: energy function can provide more flexible distribution than conditionals with known forms, such as Gaussian. For negative samples, they used SGLD-based sampling, and the resultant ELBO approximation becomes tractable and efficient. They also propose a novel OOD detection metric which seems to be superior than the previous metrics. The experimental results support their claim.

**Strengths:**

- Motivation is clear: we need to model the joint P(X,Y) in order to detect and better adapt to OOD X's. Also, it's appealing that energy function can provide more expressivity than conditionals of known forms.
- Writing is mostly clear: It's well structured and I enjoyed reading this paper
- The resultant ELBO objective in (8) makes sense, and seems efficient to train.
- Experiments are extensive and sufficient to show that EBML is competitive.
- Figure 2 provides good insights as to what the role of the two energy terms is.

**Weaknesses:**

[Major comments]

- I wonder whether the techinical contribution is sufficient. It seems to me that the proposed EBML is a replacement of conditional P(Y|X) with E(X,Y), and I feel that it would be a little bit straightforward. I'm not entirely sure if there exists similar papers doing similar things, and I found a seemingly-relevant paper [1]. Could you discuss the relationship between EBML and [1]?
- In (10), I don't understand how argmin_y does make sense. The argmin completely throws away all the uncertainty informations by finding only a single local optimum. Thus I failed to understand how the uncertainty plots from the sinusoidal experiments could have been drawn. Maybe argmin makes more sense for classifications, but it's same that all the uncertainty information is discarded. Could you clarify this point?
- It seems that this argmin operation is computationally more expensive than the usual direct modeling of P(y|x). The wall-clock time analysis should be properly compared agains the baselines.
- I don't understand why (11) should be seen as "adapting to ID". To me, the goal of (11) is to adapt $q_\psi$ to the OOD task given at the meta-test time, no?

[Minor comments]
- In (5), it would be good to explain why $q_\psi$ can be only conditioned on $\mathcal{T}^s_i$ rather than the entire $\mathcal{T}_i$, based on ABML paper (for the readers who are not familiar with the context).
- In section 4.1, it would be good to explicitly mention that the meta-parameters are collected into $\theta = \{\psi, \omega, \lambda\}$
- I my personal experience, energy functions are not always easy to train. I wonder how much it is easy to train the energy functions you proposed with wall-clock convergence plots (compared to the usual directed models P(Y|X)).
- In L179-183, I think the other methods can also use the ELBO approximation, and there will be no such limitations, no?
- In L190, maybe "only a single forward pass" rather than "only forward pass"?

[Reference]
[1] Willette et al, Meta-Learning Low Rank Covariance Factors for Energy-Based Deterministic Uncertainty, ICLR 2022

**Questions:**

See the comments above.
I'm willing to increase my score if the above major concerns (and hopefully minor comments as well) can be resolved.

**Limitations:**

Authors did properly addressed several limitations of their paper.

---

> ### Author Rebuttal · Authors · 2023-08-10
>
> We sincerely appreciate your comments on this paper. You may find our response below for your major and minor concerns. We would appreciate it if you could let us know if you have any further concerns.
> ##### Technical contribution
> > - First, we respectfully point out the misunderstandings in the argument on that EBML is a straightforward replacement of conditional $P(\bf Y|X)$ with $E(\bf X,Y)$.
> >   - As in Line 134-135, the objective of EBML is to model $P(\bf X,Y)$ instead of $P(\bf Y|X)$, thereby **resolving the issue of incomplete OOD coverage**  (c.f. Line 32-39 and Fig. 1(a)).
> >   - This, **is not as straightforward as building $E(\bf X,Y)$**, as $(\bf X, Y)$ representing a task is **a variable set of samples**. Thus,
> >     - different from EBMs conventionally applied on samples **in fixed dimension**, the energy-based modelling for $P(\bf X,Y)$ models **the distribution over sets**, which requires **(1)** flexibility in handling sets in varying cardinalities, and **(2)** expressiveness in capturing the set-level property of a task, including the function $p(y|x)$ and sample variance of a task.
> >     - the straightforward energy-based modelling strategies that meet the above first requirement unfortunately fail to meet the second, thereby being inadequate. E.g., building an EBM on all support samples of all tasks cannot even describe the function of each task; an EBM over the task representation via simple average of all samples in a task disregards sample variance. We show these simple alternatives significantly underperform EBML empirically in Table r2 (in global response PDF).
> >     - EBML directly derived from the task distribution $P(\bf X,Y)$ (c.f. Eqn. (5)) resorts **two novel EBMs** to fulfill the above two requirements, respectively: **(1)** the latent prior EBM is an energy function of the **prior** that functionally characterizes a task in the **fixed latent representation space**, correlated with **the functional distance of a task from the ID distribution**; **(2)** the task-specific data EBM is an energy function of **the observed support set conditioned on its prior**, correlated with **the variance of samples in a task** (see Fig. 2 left).
> > - Second, our technical contributions include more than modelling $P(\bf X,Y)$ with energy-based models only. Others include:
> >   - the **Energy Sum** as a principled (proportional to the likelihood of a task (c.f. Appendix A.2)) and effective OOD task detection score for EBML,
> >   - the **OOD task adaptation strategy in Eqn. (11)** to improve meta-generalization for EBML which makes it a coherent framework.
> > - Third, to our best knowledge, EBML is the first **coherent probabilistic model** that allows both detection and adaptation of OOD tasks and **general framework** being compatible with off-the-shelf meta-learning backbones. Still, we sincerely thank the reviewer for pointing out the related work of [1] and will definitely discuss the key differences between ours and [1] in the final version of the manuscript. **Please kindly refer to our global rebuttal for a detailed analysis between EBML and [1].**
> >
> > [1] Meta Learning Low Rank Covariance Factors for Energy Based Deterministic Uncertainty, ICLR 2022.
> ##### Explanation of $\text{argmin}_y$ in Eqn. (10) and the uncertainty plots for sinusoids
> > Given (1) the support set  $\mathcal{T}^s$ and (2) a query input $\mathbf{x}^q_j$ of a meta-testing task, our probabilistic model $\mathbb{E}\_{\phi\sim q_\psi(\phi|\mathcal{T}^s)}[E_{\omega}(y;\mathbf{x}\_{j}^q,\phi)] \propto -\log p(y|\mathcal{T}^s,\mathbf{x}\_{j}^q)$ allows for evaluation **at any value of $y$**. Therefore,
> > - The $\text{argmin}\_{y}$ is for finding **the most likely** prediction which is **the single desired output** in practice. This is also a common practice in making predictions by EBMs [r1].
> > - The $\text{argmin}\_{y}$ operation **does not alter the probabilistic nature** of our energy-based models, which suffices to estimate the uncertainty at $\mathbb{E}\_{\phi \sim q_\psi(\phi | \mathcal{T}^s)}[E_{\omega}(\mathbf{x}\_{j}^q, y, \phi)]$ by specifying $y$ together with $\mathcal{T}\^s$ and $\mathbf{x}\^q_j$. Note that the way that EBM offers uncertainty, **requiring specification and input of a spectrum of $y$ values**, is different from conventional probabilistic models (e.g., with Gaussian priors) that take only $\mathbf{x}$ as input and output the variance as uncertainty.
> > - That said, to draw the predictive distribution for a sinusoid task in Fig. 1 and 4 , we (1) infer $\phi\sim q_\psi(\phi|\mathcal{T}^s)$ , (2) evaluate $E_{\omega}(\mathbf{x},y,\phi)$ over a **2D grid of $x,y$**. Similar visualizations are also present in the literature (c.f. Fig. 2 in [r1]).
> > [r1] Energy-Based Models for Deep Probabilistic Regression, ECCV, 2020.
> ##### In L179-183, I think the other methods can also use the ELBO approximation, and there will be no such limitations, no?
> > - We clarify that Line 179-183 address the limitations in using naïve Monte Carlo approximation for the intractable log-likelihood of a task as a density-based OOD score. These limitations motivate us to use the ELBO approximation which leads to the proposed Energy Sum.
> > - We have investigated the possibility of using the ELBO approximation as the OOD score in a similar fashion to Energy Sum for ABML and CNPs in our ablation studies in Table 3, i.e., "SNLL + Gauss Prior".
> > - Our proposed Energy Sum outperforms these alternatives which illustrates the advantages of the proposed EBML in overcoming the issue of **limited expressiveness** of traditional probabilistic methods.
> ##### Due to the space limitation, **please kindly refer to our global rebuttal for response to the reviewer's comments** on (1) Why (11) should be seen as "adapting to ID"; the goal of (11) is to adapt $q_\psi$ to the OOD task, (2) wall-clock convergence plots (3) other minor comments

---

> > ### Comment · Reviewer_92KR · 2023-08-15
> > **Thanks for clarification**
> >
> > Dear authors,
> >
> > I appreciate all the clarifications you made during the rebuttal period. I'm satisfied with the response and hence raise my score to 7.

---

> > > ### Author Response · Authors · 2023-08-15
> > > **Thank you**
> > >
> > > Thank you very much for the positive feedback and increasing the score! We will follow your suggestions in the final version.
> > > Best, Authors.

---

### Official Review · Reviewer_u8ca · 2023-07-19

**Soundness:** 4 excellent
**Presentation:** 4 excellent
**Contribution:** 4 excellent
**Rating:** 7
**Confidence:** 3

**Summary:**

The paper deals with the problem of detecting and adaptation of out-of-distribution (OOD) tasks in the meta-learning algorithms. Recent solutions adapt Bayesian meta-learning methods which have certain limitations in terms of complete coverage of OOD tasks and based on known probability distribution that may not express complex probabilistic structure of the meta-training task distribution. Moreover, cross-domain meta-learning algorithms may adapt the OOD tasks but they are not a good estimator for OOD tasks detector. Given these limitations, the authors propose an Energy-Based Meta-Learning (EBML) coherent framework that covers both the OOD task detection and the adaptation of OOD tasks during the meta-testing. The approach is agnostic to the meta-learning backbones and can be fit with any approach to make them generalizable against OOD tasks. Experiments on the regression and classification datasets show the efficacy of the approach over the Bayesian meta-learning methods and cross-domain meta-learning approaches for OOD task detection and adaptation respectively.

**Strengths:**

-	The paper proposes a coherent energy based meta-learning framework that covers both OOD tasks detection and adaptation to alleviate the limitations of Bayesian meta-learning algorithms and cross-domain meta-learning algorithms
-	Proposes task-specific data EBM and latent prior EBM leveraging from [1]

[1] Learning Latent Space Energy-Based Prior Model, NeurIPS 2022

-	Theoretical derivation of EBML meta-learning objective with both task specific $E_{\omega}$ and latent prior $E_{\lambda}$ EBM. The author leverages both these EBMs $E_{\omega}$ and $E_{\lambda}$ to define energy score for OOD tasks detection.
-	Extensive experimental and ablation study including EBM prior v/s Gaussian for energy sum to achieve better OOD detection, more baselines for OOD adaptation, reliability of current cross-domain meta-learning algorithms under insufficient support set datapoints.
-	EBML outperforms Bayesian meta-learning approaches for OOD task detection with an improvement of up to 7% on AUROC. Compared to cross-domain meta-learning algorithms for OOD tasks adaptations on Meta-dataset, it performs on-par with the SOTA approaches.


**Weaknesses:**

- Can the authors please shed light on computation complexity of the method? It looks a good trade-off for a synthetic regression tasks, but for a real meta-dataset (classification task), it is more expensive than SimpleCNAP. Given that what approach would be have better trade-off given the performance and training time?


- The approach performs better on dataset like MNIST in meta-dataset, but for the other datasets like VGG Flower, Oxford, and MSCOCO, the results are on par with TSA. Could the authors provide the performance of the approach by plugging it into other meta-learning algorithms of classification tasks?

- Does the model handle the catastrophic forgetting about the in-distribution meta-training tasks?

**Questions:**

Please see questions in the weakness section

**Limitations:**

The authors have adequately addressed the limitations of the paper.

---

> ### Author Rebuttal · Authors · 2023-08-10
>
> We sincerely appreciate your positive feedback and comments to improve our paper. We detail our response below point by point. Please kindly let us know if our response addresses the issues you raised in this paper.
> ##### Additional results for EBML on meta-datasets.
> > We will include URL [31] (which is currently the official second-best method on Meta-dataset; note that in our experiments we have already included TSA [30] which is the best) and EBML-URL as additional baselines and experimental results in Table 4.
> > - For URL we use the official code which is publicly available. To implement EBML-URL for task-specific adaptation, we optimize Eqn. (11) w.r.t. to the task-specific feature projection matrix in URL.
> > - We show the results of URL vs. EBML-URL, together with SimpleCNAPs vs EBML-SimpleCNAPs, **in Table r1 in the PDF file attached to our global response to reviewers**, from which we observe that EBML-URL outperforms URL on 4/5, and EBML-SimpleCNPAs outperforms SimpleCNAPs 5/5 OOD datasets and conclude with the efficacy of the proposed EBML.
> ##### On performance vs computation complexity trade-off of EBML-SimpleCNAPs  for meta-dataset (classification task).
> > - Based on the OOD tasks detection results in Table 2 and additional OOD tasks adaptation results in Table r1 in our global response to reviewers, we believe that EBML-SimpleCNAPs also shows a reasonable performance vs complexity trade-off on meta-datasets:
> >   - In Table 2, EBML-SimpleCNAPs with energy sum improves the OOD detection performance of simpleCNAPs with conventional OOD detection methods by at least 7% in AUROC, 5% in AUPR and **a significant 36% reduction in FPR95**.
> >   - In Table r1, we optimize Eqn. (11) to adapt the task-encoder hence the task-specific FiLM parameters in SimpleCNAPs  for OOD meta-testing tasks in Meta-dataset. The resultant **EBML-SimpleCNAPs  outperforms SimpleCNAPs on all 5/5 OOD datasets**.
> >  - Given that we observe a noticeable gain in performance over the baseline method, we think that the computational overhead for EBML-simpelCNAPs in Appendix C.4. is therefore reasonable.
> ##### On catastrophic forgetting on the in-distribution meta-training tasks.
> >- We think that catastrophic forgetting on the ID meta-training tasks may be less of a concern in the standard meta-training and meta-testing settings due to the reasons below.
> > - However, we are not so sure whether we have correctly interpreted the review's view on "catastrophic forgetting on the ID meta-training tasks" when answering this question. Therefore, we wish the the reviewer could kindly elaborate more on this question if our answer is not sufficient.
> >
> >   - Given a meta-testing task, meta-learning algorithms always perform task-specific adaptation (i.e., the inner-loop optimization w.r.t. the model) on the support set before making prediction on the query samples.
> >   - The task-specific model is always reset back to the original state i.e., the meta-learned model on IID meta-training tasks, at the start of meta-testing on each new test task. Thus the adaptation and testing for different **meta-testing tasks are treated as independent episodes**, i.e., the task-specific model for one task will not be carried on for adaptation and testing for another different task.
> >- Since, the **meta-learned model initialization for meta-testing is the optimal model returned from meta-training on IID tasks**, therefore, there should be no forgetting on the ID meta-training tasks when testing on ID meta-training tasks.

---

> > ### Comment · Reviewer_u8ca · 2023-08-21
> > **Thanks for the response**
> >
> > I thank the authors for their detailed rebuttal.
> >
> > * By looking at the new results on meta-dataset by plugging the approach in URL [31], it looks like the performance gain in terms of mean accuracy is marginal and given the confidence interval, it could become negligible.
> >
> > * The authors showed the improvement of the OOD tasks detection in the few-shot regression and based on that results, EBML-SimpleCNAPs also show a reasonable performance vs complexity trade-off. However, given the marginal improvement in the OOD task adaptation (classification), the above statement might be not true at all. Does it mean that the approach is more suitable for OOD task adaptation?
> >
> > * I appreciate the clarifications on the catastrophic forgetting. I'm happy with the response.

---

> > > ### Author Response · Authors · 2023-08-22
> > >
> > > We appreciate the reviewer for raising your remaining concern. Please find our response below, and kindly let us know if it satisfactorily addresses your concern.
> > >
> > > - First, as stated in the Introduction, our approach aims to secure meta-generalization **within a coherent framework** by taking a **two-step** approach, i.e., (1) detection of OOD tasks and (2) adaptation of OOD tasks. Therefore,
> > >
> > >   - The evaluation of EBML, along with the baselines, grounds on the **combined performance** of both detection and adaptation aspects, instead of considering each separately.
> > >
> > >   - The performance improvement of EBML-SimpleCNAPs over SimpleCNAPs is significant, **combining** (1) **36% reduction in FPR95** for OOD detection and (2) **average 1% improvement in classification accuracy** for OOD adaptation.
> > >
> > > - Second, even when considering the OOD adaptation performance alone, we would like to justify that **the improvement is not trivial**.
> > >
> > >   - In Table 8 of [28],
> > >
> > >     - the average performance improvement of the SOTA method TSA over the best baseline URL on unseen tasks (i.e., OOD tasks in our work) is also **1%**;
> > >
> > >     - the performance improvement of URT, the third best method, over SimpleSNAPs on OOD tasks is even negative with **-0.2%**.
> > >
> > >   - EBML **consistently outperforms** all the baselines, with **an average improvement of 1%**.
> > >
> > >     - EBML-TSA **achieves the new SOTA**, via further improving TSA by **1%** in both versions of Meta-datasets (see Table 4 in the main text and Table 10 in Appendix C.2).
> > >
> > >     - EBML-URL and EBML-SimpleCNAPs with an average performance of **63%** and **59.06%** (improving URL and SimpleCNAPs by 0.4% and 1.4% on average, respectively) have ranked as the second/third best method (see "Average Unseen" in Table 8 of [28]).

---

### Official Review · Reviewer_Ghnn · 2023-08-01

**Soundness:** 3 good
**Presentation:** 3 good
**Contribution:** 2 fair
**Rating:** 5
**Confidence:** 3

**Summary:**

This paper addresses the problem of meta-learning on out-of-distribution (OOD) tasks and proposes a solution to improve the generalization capability of meta-learned prior knowledge in safety-critical applications. The paper introduces an energy-based coherent probabilistic model that enables both detection and adaptation of OOD tasks. The proposed Energy-Based Meta-Learning (EBML) framework outperforms state-of-the-art Bayesian meta-learning methods in OOD task detection and cross-domain meta-learning approaches in OOD task adaptation.

**Strengths:**

1. **Coherence and Generality**: The EBML framework offers a coherent model that supports both the detection and adaptation of OOD tasks, providing a general solution against OOD tasks for any off-the-shelf meta-learning approach.
2. **Experimental Support:** The paper showcases empirical validation of the EBML framework, demonstrating its superior performance over other methods on multiple regression and classification datasets.

**Weaknesses:**

While the paper presents an innovative application of energy-based models for out-of-distribution (OOD) task generalization and detection in meta-learning, it could benefit from a more detailed exploration of its unique contributions. Energy-based models are widely used in areas such as OOD generalization and detection. Therefore, distinguishing the specific merits of the Energy-Based Meta-Learning (EBML) framework would help elucidate its contribution to the field. The authors could potentially solidify their work's significance by providing an in-depth comparison with other techniques, or illustrating how their approach uniquely addresses challenges inherent to OOD tasks in meta-learning. This could illuminate the distinct importance of their work in this vibrant field.
[1] Energy-based Out-of-distribution Detection, NeurIPS
[2] Active Learning for Domain Adaptation: An Energy-based Approach, AAAI

**Questions:**

Please refer to the weakness

**Limitations:**

The paper discussed about the limitation.

---

> ### Author Rebuttal · Authors · 2023-08-10
>
> We sincerely appreciate your constructive comments to improve our paper. We detail our response below point by point. Please kindly let us know if our response addresses the questions you had for this paper.
>
> ##### On how EBML uniquely addresses the challenges inherent to OOD tasks in meta-learning.
>
> > - In conventional sample-level OOD detection, measuring the OODness boils down to **(1)** modelling of the ID distribution and **(2)** evaluation of the likelihood that the to-detect sample belongs to the ID distribution. EBM has shown its superiority in **(1)** flexibly modelling the ID distribution and **(2)** deducing the energy score as a principled but simple OOD measure.
> >
> > - Disparate from OOD detection of samples that are **in fixed dimension**, detection of an OOD task **as a variable set of samples** poses the following **two inherent challenges**.
> >
> >   - **(1)** Modelling the ID distribution over sets requires both flexibility in handling sets in varying cardinalities and expressiveness in capturing the set-level property of a set. Thus, a direct application of EBM via naive aggregation of samples within a task, ignoring the sample variance and failing to fully meet the second requirement, is inadequate.
> >
> >   - **(2)** The likelihood of a task belonging to the ID task distribution involves comparison between a set and a set distribution and reduces to calculating the set distance which is widely accepted more complicated than sample distance.
> >
> > - To address the above two challenges, EBML uniquely
> >
> >   - **(1)** derives two novel EBMs to jointly model the ID task distribution, including a latent prior EBM and a task-specific data EBM.
> >
> >     - Latent prior EBM is an energy function of the prior, which describes a task in the **fixed latent representation space regardless of the cardinality** and correlates with **the distance of a task from the ID distribution** (see Fig. 2 right).
> >
> >     - Task-specific data EBM is an energy function of the observed support set conditioned on its prior, correlated with **the variance of samples in a task** (see Fig. 2 left).
> >
> >   - **(2)** introduces the Energy Sum as an effective OOD task detection score, which
> >
> >     - has been proved exactly **proportional to the likelihood of a task** (see Appendix A.2),
> >
> >     - evaluates the above two aspects, i.e., distance and variance, and
> >
> >     - enjoys empirical **superiority over the scores for OOD sample detection** (adapted by averaging scores of the samples in a task, see Table 2).
> >
> > - Furthermore, EBML is more than just an OOD task detector in meta-learning; it improves generalization to OOD tasks in meta-learning via the proposed adaptation strategy in Eqn. (11).
>
> ##### Comparison between EBML and [1,2]
>
> > - Both [1] (i.e., [35] in our main text) and [2] investigate the application of EBMs in **either** OOD sample detection **or** active domain adaptation, so that they do not face the aforementioned two challenges **inherent to OOD tasks in meta-learning**.
> >
> > - We will definitely follow the reviewer's great suggestion to improve the paragraph of "EBMs for OOD Detection" in Section 2, by
> >
> >   - detailing the abovementioned unique contributions of EBML in detecting OOD tasks,
> >
> >   - including [2] which first queries unlabeled target samples by the difference between the minimum and the second minimum energies and then aligns the free energies of selected target samples with those of source ones. Thus, the EBM in [2] still models the distribution over **samples** for sampling, instead of over **sets** in our work.
> >
> > [1] Energy-based Out-of-distribution Detection, NeurIPS 2020
> > [2] Active Learning for Domain Adaptation: An Energy-based Approach, AAAI 2022

---

### Author Rebuttal · Authors · 2023-08-10

### Global Response to Reviewers
#####  Why (11) should be seen as "adapting to ID"; the goal of (11) is to adapt $q_\psi$ to the OOD task
> As we explain below, our original description of Eqn. (11) in Line 221, i.e., "adapts this inadequate meta-learned prior back to the ID region" **is not conflicting with** reviewer 92kR's interpretation that Eqn. (11) is "adapting $q_\psi$ to the OOD task". Nevertheless, we thank the reviewer for pointing this out, and we will make sure to add more explanation on this in section 4.3 to avoid confusion in the final manuscript.
> - As shown in Eqn. (11), the **optimization procedure** is indeed with respect to the mapping $\zeta$ that adapt the parameters $\psi $ of $q_\psi$ to $\psi\cup\zeta$, so that $q_{\psi\cup\zeta}$ accommodates the OOD task.
> - The **optimization result**, however, is that the resulting OOD prior $\phi\sim q_{\psi\cup\zeta}(\phi|\mathcal{T}^s)$ approaches the ID region where the meta-generalization is guaranteed due to meta-training on ID tasks. We illustrate the adaptation process in Fig. 3, where the OOD priors $\phi$ indeed shift towards the ID region as optimization proceeds.
> - We provide an analogy with classifier editing [48] in Line 222, in which the classifier is edited/adapted to accommodate an OOD image (a car with wooden tire) so that the predicted class of the OOD image aligns with an ID one (a car with rubber tire).

##### We sincerely thank reviewer 92KR for pointing out the related work of [1] and will definitely include the following key differences between ours and [1] in the final version of the manuscript.
>   - **Problem definition and objective:** EBML aims to detect a meta-testing task that is OOD of the meta-training tasks; however, [1] focuses on detecting a query sample that is OOD of the support samples in a meta-testing task.
>   - **Method:** EBML falls into the category of **density-based** OOD detection methods, so that it **explicitly meta-learns the distribution of meta-training tasks via the two proposed EBMs** and develops the **Energy Sum to flag those high-energy tasks as OOD tasks**; however, [1] first meta-learns class covariance matrices as a parameterised function to alleviate the collapse of the covariance matrix due to limited samples in a few-shot meta-testing task and secondly resorts to **post-hoc** OOD detection via **energy scaling**. Note that "energy scaling" here refers to temperature scaling in softmax output, **without learning any EBM**.
>   - **Coherence and Generality**: EBML also improves meta-generalization, and works well in either regression/classification; however, [1] builds on only generative classifier-based meta-learning approaches solves only classification tasks and underperforms some very basic meta-learning algorithms e.g., MAML, ProtoNet, in classification accuracy (c.f. Tables 2,3,4,5 in the Appendix of [1])
>
>     | Method | ID                                     | OOD to evaluate     | OOD detection method | EBM explicitly learned | Coherency                                                  | Generality                                                 |
>     | ------ | -------------------------------------- | ------------------- |:-------------------- | ---------------------- | ---------------------------------------------------------- | ---------------------------------------------------------- |
>     | EBML   | meta-training tasks                    | a meta-testing task | Density-based        | Yes                    | both OOD detection and adaptation                          | all meta-learning backbones, regression and classification |
>     | [1]    | support samples in a meta-testing task | a query sample      | Post-hoc scaling     | No                     | uncertainty calibration with a sacrifice in generalization | generative-classifier based backbones, classification      |
>
>     [1] Meta Learning Low Rank Covariance Factors for Energy Based Deterministic Uncertainty, ICLR 2022.

##### Wall-clock time analysis and convergence plots for EBML, and additional results for OOD tasks adaptation for Meta-datasets.
> - Please kindly refer to the additional plots and tables in the PDF file attached to this global response.

##### Other minor comments related to writing
> - We sincerely thank the reviewers for their helpful suggestions that would further improve the clarity of the paper.  We will make sure to follow these comments when editing the final version of the manuscript

---

### Decision · Program_Chairs · 2023-09-21

**Decision:**

Accept (poster)

**Comment:**

All reviewers unanimously support the acceptance of the paper. Please incorporate the feedback from the rebuttal in the camera ready.

Congratulations!